# Detecting the Hadron-Quark Phase Transition with Gravitational Waves

**Matthias Hanauske** [1,2,*], **Luke Bovard** [1,2], **Elias Most** [1], **Jens Papenfort** [1], **Jan Steinheimer** [2], **Anton Motornenko** [1,2], **Volodymyr Vovchenko** [1,2], **Veronica Dexheimer** [3], **Stefan Schramm** [2] and **Horst Stöcker** [1,2,4]

1   Institute for Theoretical Physics, Goethe University, Max-von-Laue-Straße, 1, 60438 Frankfurt am Main, Germany; bovard@th.physik.uni-frankfurt.de (L.B.); most@fias.uni-frankfurt.de (E.M.); papenfort@itp.uni-frankfurt.de (J.P.); motornenko@fias.uni-frankfurt.de (A.M.); vovchenko@fias.uni-frankfurt.de (V.V.); stoecker@fias.uni-frankfurt.de (H.S.)
2   Frankfurt Institute for Advanced Studies, Ruth-Moufang-Straße, 1, 60438 Frankfurt am Main, Germany; steinheimer@fias.uni-frankfurt.de (J.S.); schramm@fias.uni-frankfurt.de (S.S.)
3   Department of Physics, Kent State University, Kent, OH 44243, USA; vdexheim@kent.edu
4   GSI Helmholtzzentrum für Schwerionenforschung GmbH, 64291 Darmstadt, Germany
*   Correspondence: hanauske@fias.uni-frankfurt.de; Tel.: +49-69-798-47869

**Abstract:** The long-awaited detection of a gravitational wave from the merger of a binary neutron star in August 2017 (GW170817) marks the beginning of the new field of multi-messenger gravitational wave astronomy. By exploiting the extracted tidal deformations of the two neutron stars from the late inspiral phase of GW170817, it is now possible to constrain several global properties of the equation of state of neutron star matter. However, the most interesting part of the high density and temperature regime of the equation of state is solely imprinted in the post-merger gravitational wave emission from the remnant hypermassive/supramassive neutron star. This regime was not observed in GW170817, but will possibly be detected in forthcoming events within the current observing run of the LIGO/VIRGO collaboration. Numerous numerical-relativity simulations of merging neutron star binaries have been performed during the last decades, and the emitted gravitational wave profiles and the interior structure of the generated remnants have been analysed in detail. The consequences of a potential appearance of a hadron-quark phase transition in the interior region of the produced hypermassive neutron star and the evolution of its underlying matter in the phase diagram of quantum cromo dynamics will be in the focus of this article. It will be shown that the different density/temperature regions of the equation of state can be severely constrained by a measurement of the spectral properties of the emitted post-merger gravitational wave signal from a future binary compact star merger event.

**Keywords:** equation of state; hadron-quark phase transition; binary neutron star merger; gravitational wave

## 1. Introduction

When Albert Einstein in 1915 presented his General Theory of Relativity (GR) to the scientific community, not many physicists believed that his new theory of space–time deformation could replace the old Newtonian viewpoint of gravitation. Today, just over a hundred years later, his revolutionary new theory of gravitation has passed all experimental and observational tests. The well known *Einstein Equation* of GR formulates theoretically the mathematical basis for gravitational waves (GWs) and black holes (BHs). Recently, a little more than a hundred years after the birth of GR, the first picture

of a BH had been observed in millimeter-wavelength by the *Event Horizon Telescope Collaboration* [1]. Simulations performed by the *EHT Collaboration* show that the picture could be explained accurately by numerical results of accretion disks around a rotating Kerr BH [2,3]. This cornerstone in the history of astronomy is only one example of the success of GR. Within this article we will focus on another milestones achieved within the last years.

On 14 September 2015, almost exactly a hundred years after Albert Einstein developed the field equations of GR and predicted the existence of GWs, these curious spacetime-ripples have been observed from a pair of merging black holes by the LIGO detectors (GW150914, [4]). By means of the gravitational-wave detectors LIGO and VIRGO, 11 gravitational waves have now be detected [5], whereby one of these GWs (GW170817) was caused by the collision of two neutron stars [6] about 130 million years ago. Electromagnetic radiation in all frequency ranges was also detected during this event [7,8] and an emitted gamma-ray burst (GRB 170817A, [9]) hit the gamma-ray satellite telescopes with a delay of 1.7 s. Space-based gamma-ray telescopes (e.g., the Fermi's gamma-ray burst monitor or the Swift gamma-ray burst mission) detect on average approximately one gamma-ray burst per day—however, the gamma-ray burst [9] that had been associated with GW170817 is an outstanding event and, in addition with the observations of the electromagnetic counterparts of the associated kilonova, provides a conclusive picture of the whole merger event. This coincidence of the direct detection of a GW from a neutron star collision with the emitted short gamma-ray burst was the first observational proof that binary neutron star (BNS) mergers generate short gamma-ray bursts.

Due to the lower sensitivity of the GW detectors in the high frequency range, only the inspiral phase of GW170817 had been observed. The inspiral phase of a BNS merger and the emitted GW waveforms before merger can be described by approximate analytic solutions of the Einstein field equations using the post-Newtonian approach [10,11] or effective-one-body calculations [12,13]. However, the understanding of the conjunction of the GW170817 event with the emitted gamma-ray burst GRB170817A and its associated kilonova, and the prediction of the post-merger gravitational wave profiles from BNS mergers are dominated by the results of numerical simulations, which solve the hydrodynamic evolution of elementary dense and hot matter on a curved space–time grid using the field equations of GR. Taking the results of various BNS merger simulations into account, the overall picture of the GW170817 event can be regarded as the following sequential chronology: Right after the neutron stars merge, a remnant is formed, which could be stable for longer times (supramassive neutron star, SMNS) or collapse within less than one second (hypermassive neutron star, HMNS) to a rotating Kerr black hole. During this process, a short gamma ray burst is emitted, releasing in less than one second the energy emitted by our Galaxy over one year [14]. A large number of numerical-relativity simulations of merging neutron star binaries have been performed over the last 15 years and the emitted GWs and the interior structure of the generated SMNS/HMNS have been analyzed in detail. Based on these large number of numerical-relativity simulations, the emitted GWs, the interior structure of the generated HMNS/SMNS, the accurate measurement of the amount of ejected material from the merger, the synthetic light curves of the produced kilonova signal, the distribution of the abundances of heavy-elements, and last but not least, the impact of magnetic fields on the long term ejection of mass have been investigated in detail. The observed kilonova of the follow-up observations of GW170817 [7] provided the first definitive and undisputed confirmation of a kilonova and the formation of r-process elements from merging neutron stars. To investigate the r-process formation in merging neutron stars, a variety of simulations were performed using numerous equations of state (EOSs), initial masses, and mass ratios that sample well the parameter space of the GW170817 event [15,16]. Finally, the observed tidal deformations of the two neutron stars from the late inspiral phase and other properties of GW170817 constrained severely the equation of state (EOS) of dense matter (see Section 2).

This article is structured as follows: Section 2 summarizes the main findings of the long-awaited event GW170817. The temperature and density structure of a neutron star merger product will be analysed in Section 3 (for details see [17–22]). Similar to [23], the different phases of a BNS merger

scenario will be explained using the analogy of a variety of different dances and a new way of presenting the results will be used, which visualizes the evolution of hot and dense matter inside the remnant in a temperature-density ($T$-$\rho$) QCD phase diagram. It will be illustrated that the temperature and density values reached inside the HMNS requires an incorporation of a hadron-quark phase transition (HQPT) in the EOS. The different phases of a binary hybrid star merger and their connection with the emitted GW signal will be in the focus of Section 4. Section 5 is dedicated to give a summary of our results and an outlook.

## 2. GW170817—The Long-Awaited Event

In August 2017, the GWs and electromagnetic counterparts from the merger of a BNS merger were detected by the LIGO/VIRGO collaboration and numerous observatories around the world. This long-awaited event (GW170817) marks the beginning of the new field of multi-messenger gravitational wave astronomy. GW170817 provides the first direct evidence of a link between BNS mergers and short $\gamma$-ray bursts. It additionally implies that gravitational waves travel at the speed of light, with deviations smaller than a few $10^{-15}$. The detected GW in combination with the information from the observed electromagnetic counterpart constrain alternative theories of gravity, possible effects due to large extra dimensions, and dark energy models [24–26].

With the use of the observed tidal deformations of the two neutron stars from the late inspiral phase and other properties of GW170817, the EOS of dense matter could be severely constrained [27–36]. It is impressive that with only one observed BNS merger event the EOS of neutron star matter and the properties of neutron stars could be constrained in a way which was not possible two years ago. The knowledge about neutron stars and the global properties of the EOS of dense matter has grown tremendously within the last two years and the scientific literature dealing with GW170817 has increased rapidly. New constraints on the neutron star radii have been found [27,37–39] and the impact of a HQPT on these estimates were discussed [40]. An upper bound for the maximum mass of neutron stars was estimated, almost contemporaneously, by several groups [28,29], and the global properties of a HQPT using a hybrid star EOS could be constrained [31,40].

All of these calculations and estimates base on the extracted GW signal of the late inspiral phase, where the two neutron stars get tidally deformed but still have not merged to become a single remnant. At this moment, the density values reached in the interior of the stars are most likely below the onset of the HQPT (for most EOSs $\rho_{\mathrm{inspiral}} < 3\,\rho_0$, where $\rho_0 := 2.705 \times 10^{14}$ g/cm$^3$ is the normal nuclear-matter rest-mass density) and the thermal effects on the EOS and emitted wave form can be neglected ($T \approx 0$). The ultra-high density and temperature regime of the EOS, which is solely imprinted in the merger and post-merger GW emission from the remnant star (HMNS or SMNS), was not observed in GW170817, but will possibly be detected in forthcoming merger events within the current observing run [41]. As a result, the most interesting part of the high density and temperature regime of the EOS is still poorly constrained and most of the calculations of the presented constraints on the EOS do not carefully implement the possibility of a strong HQPT in their estimates. In order to understand which regime of the EOS could be constrained by a future GW detection of a BNS merger event, the interior temperature and density structure of a neutron star merger product and the evolution of the hot and dense matter inside the HMNS will be analyzed and visualized in the next section. Following [23] and former publications [17–22], the different phases of a BNS merger scenario will be explained using the analogy of a variety of different dances. The focus within this article will be on the transition from from the violent early post-merger phase (Disco-fox phase) to the middle part of the post-merger phase (Merengue phase).

## 3. Hypermassive Neutron Stars and the QCD Phase Diagram

In order to understand the post-merger evolution of a BNS merger, numerous numerical-relativity simulations have been performed in the last decades by various groups [42,43]. The emitted GW spectrum [44–47], the impact of initial spin [48–50], eccentricity [51,52], and mass ratio [53]

have been analysed in detail. The merger event is an extremely disruptive process, nevertheless, the amount of ejected material and the synthetic light curves of the produced kilonova signal have been studied [16,54–56]. It was found that the lifetime of a HMNS depends strongly on the total mass of the binary and the EOS [36,57]. Additionally, viscosity effects might be important [20,58] in this context. The stability of a HMNS is mainly caused by the rotation profile of the differentially rotating HMNS [17,49,59], which is deeply connected with the temperature structure inside the HMNS.

In the following, the results of a numerical simulation will be presented (for details see [17–22]), which had been performed in full general relativity using the `Einstein Toolkit` [60] where a fourth-order finite-differencing code `McLachlan` [61,62] had been used. The code is based on the `BSSNOK` conformal traceless formulation of the Einstein equations [63–65] using a "1 + log" slicing condition and a "Gamma-driver" shift condition [66,67]. The covariant conservation of energy, momentum, and rest mass, formulated within the general-relativistic hydrodynamics equations [68], are cast in the conservative Valencia formulation [69]. The evolution of these hydrodynamics equations is done using the `WhiskyTHC` code [70,71]. A numerical grid with an mesh refinement approach based on the `Carpet` mesh-refinement driver [72] is used to both increase resolution and extend the spatial domain. A purely hadronic model of the EOS (`LS220`, [73]) had been used and an equal-mass binary with a mass of $1.35 M_\odot$ for each star was selected. However, the results of other purely hadronic EOSs and/or different initial conditions show qualitatively a similar behaviour, as described below.

The different phases of a BNS merger can be regarded as a great love story and illustrated using a sequence of different ballroom dances (for details see [23]), ranging from the first moment of perception, the closer approach, the moment of coalescence, and fusion. So far, only the inspiral phase (Viennese waltz phase), where the two stars are separated by a certain distance and orbit around each other, has been observed in GW170817. In this inspiral phase, the temperature effects can be neglected to a good approximation. During the last orbits, the stars become tidally deformed and the temperature in the low density regime, between the centers of the two stars, increases rapidly [21–23]. At merger time ($t = 0$ ms), where the emitted gravitational wave of the newly born remnant reaches its maximum value, the temperature maximum of the newly born remnant could reach values up to 80 MeV.

Figures 1 and 2 visualize the HMNS properties right after merger within a time segment ($0.2 < t < 1.7$ ms) which corresponds to the early post-merger phase. The pictures in the left column show the spatial distribution of the rest-mass density in the equatorial plane of the HMNS, while the middle column visualizes the temperature distribution.

The pictures in the right column depicts the density-temperature profiles inside the inner region of the HMNS in the style of a ($T$-$\rho$) QCD phase diagram plot. The color-coding indicates the radial position $r$ of the corresponding fluid element inside the HMNS. The white circles/diamonds mark the maximum values of temperature/density. Additionally, using the method of a "corotating frame", several tracer particles that remain close to the equatorial plane are visualized with black dots. The final part of the tracer flowlines for the last $\Delta t = \simeq 0.19$ ms are shown and the small black dots indicate the tracer position at the time indicated in a frame (for details see [16,74]). At $t = 0$, the tracers have been placed near to the surface of the newly born remnant and, as time evolves, the tracer particles diffuse over the entire region of the HMNS. In the following, only the tracer particles that remain in the interior of the remnant are visualized, while tracers leaving the HMNS are neglected. The violent, early post-merger phase (Disco-fox phase) is characterized by a pronounced density double-core structure and hot temperature regions which are smeared out in areas between the double-core density maxima (see Figure 1). At merger time, the center of the HMNS is highly compressed ($\rho_{max} \approx 3.3\rho_0$), the corresponding large pressure pushes the matter away from the center, and a double-core structure in the density profile appears (see Figure 1). The temperatures near to the two centers of the double-core is quite small and the high temperature values are reached in the middle of the double-cores. As a result of this dynamical process, the density and pressure values in the center of the HMNS decrease, the gravitational force again compresses the system, and the extent of the double-core shrinks (see Figure 2). The results shown that, within the early post-merger phase,

the evolution of matter is quite violent and the density of a fluid cell (tracer particle) can show large variations of order 100% on a timescale of milliseconds. As a result, significant bulk viscous dissipation may occur [20].

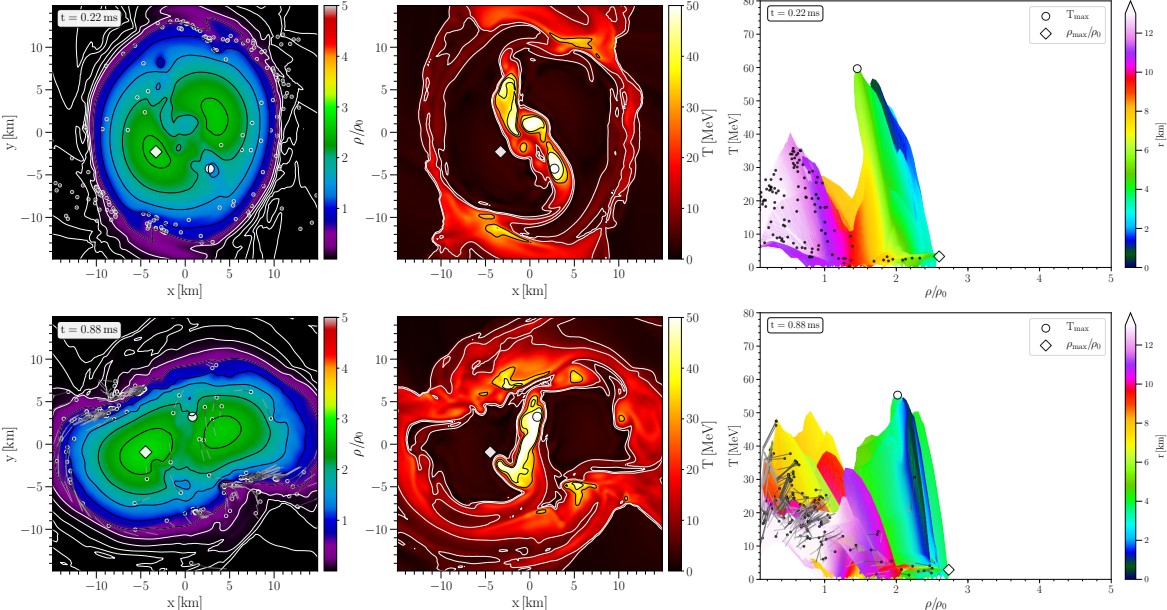

**Figure 1.** Spatial distribution of the rest-mass density (**left**) and temperature (**middle**) of the `LS220-M135` simulation at several time snapshots within the early post-merger phase. (**right**) The pictures on the right side show the density-temperature profiles inside the inner region of the HMNS in the style of a $T$-$\rho$ QCD phase diagram.

Figure 3 visualizes the HMNS properties within a time segment ($1.9 < t < 5.5$ ms), which corresponds to the transition phase from the early post-merger phase to the middle part of the post-merger phase. At $t = 1.87$ ms (upper row of pictures in Figure 3), the density double-core structure has just disappeared and the maximum density value (diamond symbol) has shifted to the central region of the HMNS, although the density distribution shows a pronounced 'peanut' shape. The hot temperature regions are still smeared out in areas between former double-core density maxima and two large regions with low temperature values are still present at the spatial allocations of the former double-core density regions.

During the first two milliseconds after merger, the high temperature and moderate density regime of the EOS is reached and the GW emitted during this early post-merger phase depends strongly on this sector of the EOS. For later times, the high temperature regions transform into two hot temperature spots and they move further out. The high temperature values ($T > 40$ MeV) are reached now in regions where the density is in a range of $1.5 - 2.5\rho_0$, while the maximum density values are always at moderate temperatures $T < 20$ MeV. The tracer particles have diffused over the entire inner region of the HMNS and populate almost the whole area of the ($T$-$\rho$) plane. Some of these tracers circulate around the high temperature hot spots, others populate the low temperature high dense inner region, and some are moving in the outer surface of the HMNS within the low density regime (see lower row of pictures in Figure 3). In [17] the deep connection between the internal rotational properties of the produced HMNS/SMNS and the evolution of the density and temperature profiles of the ultra dense and hot remnant was analysed and it was shown that the high temperature hot spots appear at the same spatial locations where a large differential rotational of the rotation profile $\Omega$ occurs.

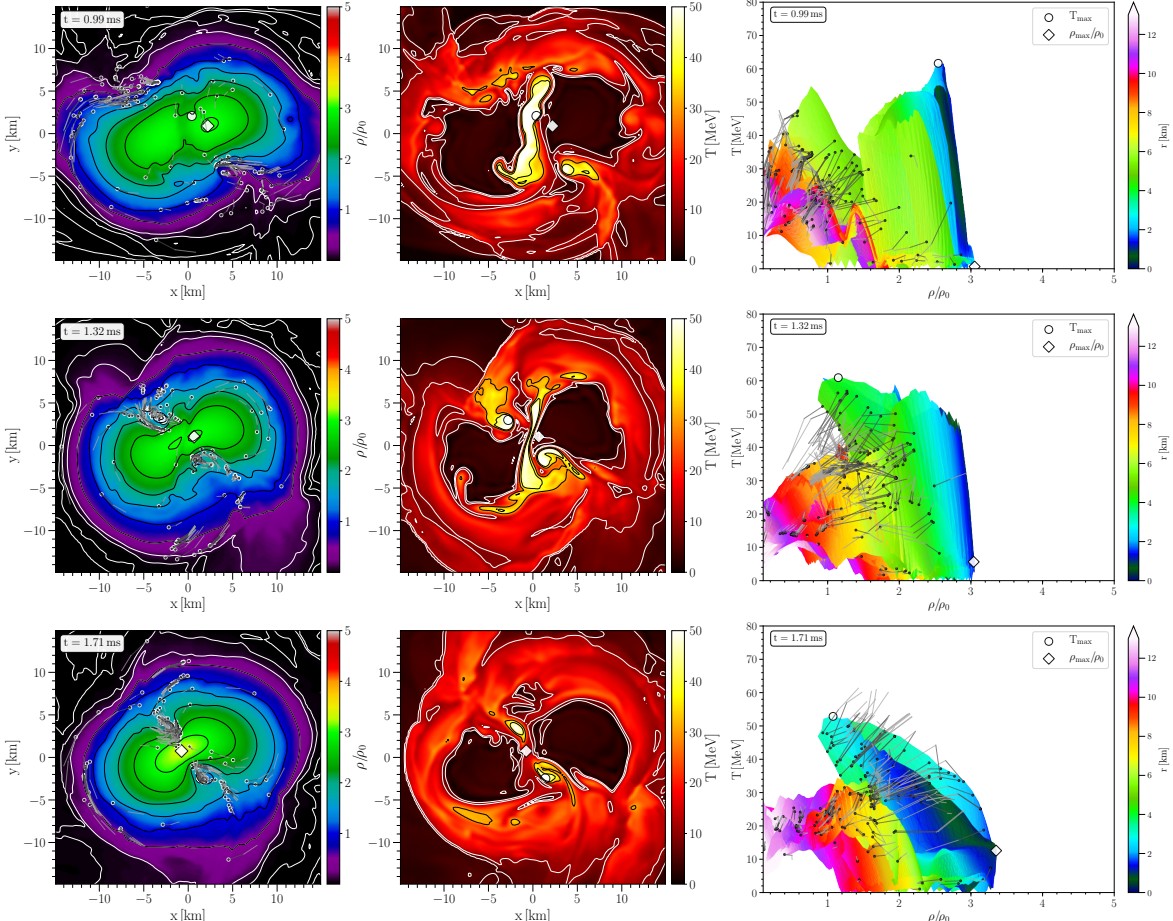

**Figure 2.** Same as Figure 1 but for later time snapshots within the early post-merger phase.

Figure 4 shows two snapshot at later times in the post-merger phase. The two pronounced temperature maxima have almost merged to one maximum. At these post-merger times, the hot spots of the temperature profile have smeared out and a ring like temperature structure emerged. Additionally, the 'peanut' shape of the density profile has been dissolved, and the area populated in the ($T$-$\rho$)-plane has been constricted to a small quasi stable region. The center of the remnant consists of extremely dense matter ($\rho/\rho_0 \approx 5$) at moderate temperature values $T \approx 10$ MeV, while the maximum of the temperature is reached at the highest point of the temperature ring like structure at moderate density values ($\rho/\rho_0 \approx 2$). The unusual temperature ring-like structure (see lower picture in Figure 4) is again closely related with the rotation profile $\Omega$ of the differentially rotating HMNS (see [17] for details).

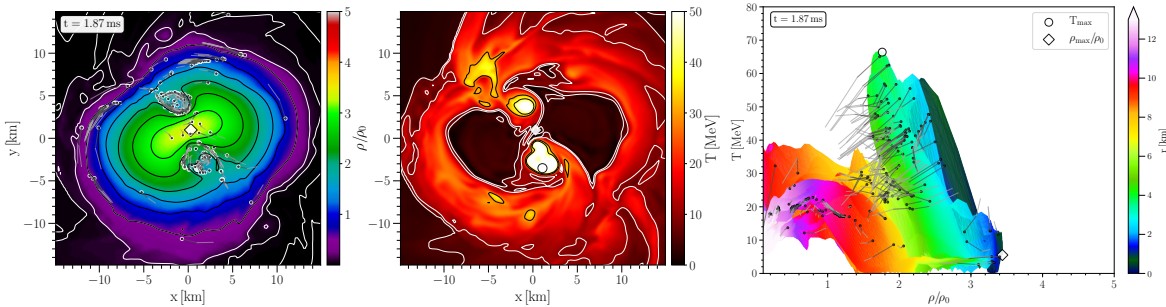

**Figure 3.** *Continued*

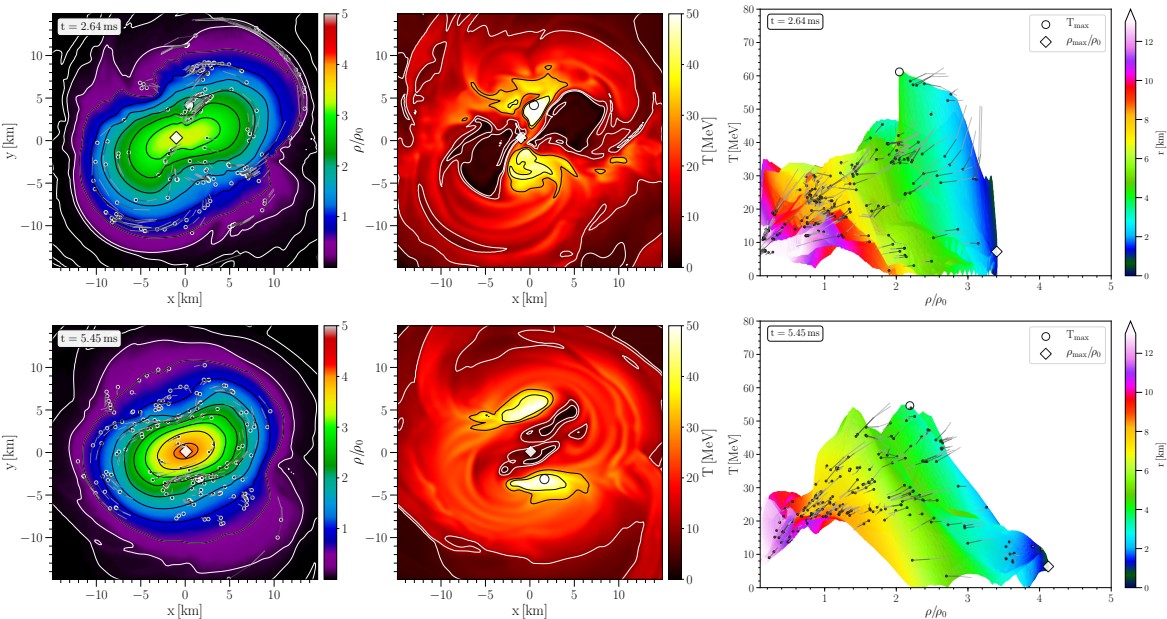

**Figure 3.** Same as Figure 2 but for later time snapshots within the transition segment from the early to the middle part of the post-merger phase.

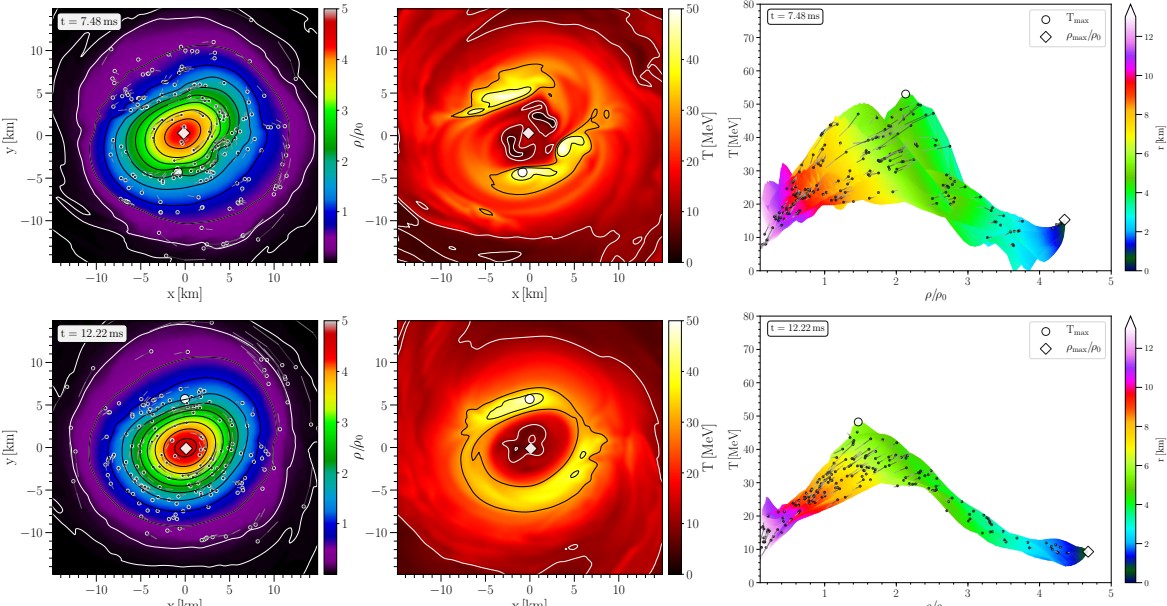

**Figure 4.** Same as Figure 3 but for later time snapshots.

## 4. Detecting the Hadron-Quark Phase Transition with Gravitational Waves

Within the former section, it was illustrated that during the post-merger evolution of the HMNS, the value of central rest-mass density increases within the LS220-M135 simulation to several times of normal nuclear matter (see Figure 4). For such high densities, the EOS is still poorly constrained by observations from heavy-ion collisions. By analyzing the power spectral density profile of the post-merger emission of a future event within the current observing run of the LIGO/VIRGO collaboration, the GW signal can set tight constraints on the high density regime of the EOS of elementary matter [75]. Numerical simulations that include a density/temperature and composition dependent EOS with a HQPT, the so called hybrid star merger simulations, have only been performed recently.

In [76] the temperature-dependent Chiral Mean Field (CMF) model [77] with a strong HQPT has been used for the first time in a BNS merger simulation. Due to the fact that the EOS does not yield gravitationally stable hybrid stars with deconfined quark matter cores (unless mixtures of phases are allowed), the effects of the HQPT can only be observed in a BNS merger scenario. In [76] it was shown that the phase transition, which happens during the post-merger phase, leads to a hot and dense quark core that, when collapses to a black hole, produces a ringdown signal different from the hadronic one. The evolution of the temperature and density in the merger remnant is different from the one generated by using a purely hadronic model (see Section 3 and Figure 3 in [76]), as the inner region of the hypermassive hybrid star forms a very hot and ultra-dense quark core before the collapse to a Kerr black hole.

In [78] the temperature-dependent, hadron-quark hybrid (DD2F-SF) model [79] has been used in a BNS merger simulation. In contrast to [77], stable hybrid stars containing both hadrons and quarks are realisable within the DD2F-SF EOS. In [78] it was shown that the dominant post-merger GW frequency $f_{\text{peak}}$ exhibits a significant deviation from the empirical relation between $f_{\text{peak}}$ and the tidal deformability $\Lambda$ if a strong first-order phase transition leads to the formation of a gravitationally stable extended quark matter core in the post-merger remnant. Such a shift of the dominant post-merger GW frequency might be revealed by future GW observations using second- and third-generation GW detectors.

Especially within an EOS that includes the possibility of a twin star behaviour, the astrophysical observables of a HQPT might be detectable in future neutron star merger events [19,21,35]. Twin stars appear when the quark EOS is stiff enough to stabilize small stars with a sizable quark core. Presupposing twin-star solutions and assuming that the hadronic part of the EOS is known up to a certain density, the global parameters of the HQPT are constrained tightly in order to explain the GW170817 event [40]. Binary hybrid star merger simulations which implement a strong HQPT including the possibility of a twin-star behaviour are currently under construction. Preliminary results show that the appearance of a strong HQPT in the interior region of the HMNS will change the spectral properties of the emitted GW. If an unstable twin-star region is reached during the post-merger evolution of the remnant, the $f_2$-frequency peak of the emitted GW signal will change due to the speed up of the differentially rotating HMNS and large twin-star oscillations might occur [19,21].

## 5. Summary and Outlook

Nearly one hundred years after Albert Einstein developed the theoretical groundings of black holes and gravitational waves, both entities have been observed. A gravitational wave event from a binary neutron star merger was detected in August 2017 by the LIGO/VIRGO collaboration (GW170817) and, with the analysis of the corresponding gravitational wave signal, the equation of state of elementary matter could be constrained severely. Recent simulations show that the appearance of a hadron to quark phase transition in the interior region of a hybrid star merger remnant might change the overall properties of the merger event and could be detectable in future.

The possible appearance of a transition from confined hadronic to deconfined quark matter and the formation of regions of deconfined quark matter in the interior of a compact star merger product have been discussed within this article. The temperature and density structure of a neutron star merger product and the evolution of hot and dense matter inside the produced hypermassive compact star advises an incorporation of a HQPT in the equation of state. The occurrence of hot temperature regions and their spatial location is closely connected with the rotational properties of the differentially rotating remnant [17]. Additionally, the possibility of a viscousless superfluid quark phase might change the overall merger properties, as viscous dissipation and energy transport can play a significant role in the survival time of the post-merger object [20]. Binary hybrid star mergers represent therefore optimal astrophysical laboratories to investigate the phase structure of QCD and in addition with the observations from heavy-ion collisions will possibly provide a conclusive picture on the QCD phase

structure at high density and temperature [18]. We discussed how the gravitational wave signals from future observations can be used to constrain the equation of state of elementary matter.

**Author Contributions:** Conceptualization, M.H., H.S.; Methodology, M.H., H.S., E.M., L.B., V.D.; Investigation, M.H., J.S., A.M., L.B., E.M., J.P., S.S., V.V., V.D.; Writing—Original Draft Preparation, M.H.; Writing—Review & Editing, M.H., V.D., H.S.; Funding Acquisition, H.S.

**Funding:** M.H. gratefully acknowledges support from the Frankfurt Institute for Advanced Studies (FIAS) and the Goethe University Frankfurt, V.D. acknowledges support from the National Science Foundation under grant PHY-1748621, while H.S. acknowledges the Judah M. Eisenberg laureatus Professur endowment.

**Acknowledgments:** We would like to thank Luciano Rezzolla. Without his profound knowledge and his comprehensive expertise in the field of numerical relativity and general relativistic hydrodynamics, the presented simulations and the whole article would not have been possible. Additionally, we would like to thank Glòria Montaña and Laura Tolos for valuable discussions.

**Conflicts of Interest:** The authors declare no conflict of interest.

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
