# Peer review of "Detecting the Hadron-Quark Phase Transition with Gravitational Waves"

_universe, doi:10.3390/universe5060156_

Reviewer 1 Report

This article discusses the thermodynamical condition of matter in neutron star mergers. The manuscript follows rather closely Refs. 18, 19, and 20, by the same authors, with the distinction that here they show a different model and some new analysis. The analysis and the discussion are interesting, and the results reported here have important implications for the understanding of matter effects in mergers. So, although they are not really new, I would still recommend the article for publication with minor changes.

I invite the authors to take the following points into account when revising the manuscript.

The abstract is a bit misleading. The authors write: "Based on a large number of numerical-relativity simulations ..." However, only one simulation is actually discussed in the article.

I fail to see the connection between the study reported in the manuscript and the results of the Event Horizon Telescope (EHT) Collaboration. The discussion of EHT results is not relevant and should probably be removed.

The choice of references in the introduction and throughout the text is quite arbitrary. The authors go through all sort of hoops to cite as many of their own papers as possible and ignore many important contributions to the field.

The manuscript lacks any description of the methods used for the simulations and the analysis. It is fine to refer to other published papers for a detailed description of the methods. However, a short summary of the methodology is needed to make the current manuscript sufficiently self contained.

Reviewer 2 Report

This paper reviews some facts about GW170817, and presents some results on

numerical simulations of binary neutron stars. Except some snapshots of the

numerical simulations, results were discussed very marginally. I believe they

should be extended to be a scientifically sound article.

- In the first paragraph, EHT has used millimeter-wavelength observations,

  instead of "radio waves".

- L51: "... gravitational wave profiles from BNS mergers are dominated by the

  results of numerical simulations"

  This is not 100% true. Actually GW170817 only had "inspiral" detection, and the

  waveform is from post-Newtonian calculations (Blanchet 2014, arXiv:1310.1528)

  and effective-one-body approach (Bohe et al. 2017, arXiv:1611.03703).

- An important result for section 4 is about the changes in f_peak and f_2. But

  only very marginal result was mentioned. I believe a bit more quantitative 

  extension is needed.

- L248: "We demonstrated how the gravitational wave signals 249 from future

  observations can be used to constrain the equation of state of elementary

  matter."

  I didn't find it "demonstrated". Only some words were mentioned that they

  **might** be detected. For demonstration, some calculation is needed to show

  that the phase transition will introduce enough changes, given the noise

  curves of detectors.

Author Response

see attached file

Round  2

Reviewer 2 Report

The authors have taken my previous comments, and modified the paper. It is suitable for publication in Universe.